# Effects of non-technical skill-based surgical education for trainees on herniorrhaphy outcomes

**Daisuke Koike** [1,2]*, **Takahiro Nishimura**[2], **Yusuke Suka**[2], **Motoki Nagai**[2], **Yukihiro Nomura**[2], **Hiroyuki Kato**[1], **Yukio Asano**[1], **Masahiro Ito**[1], **Satoshi Arakawa**[1], **Takuma Ishihara**[3], **Akihiko Horiguchi**[1]

1 Department of Gastroenterological Surgery, Fujita Health University School of Medicine, Bantane Hospital, Nagoya, Japan, 2 Department of Surgery, Asahi General Hospital, Asahi, Chiba, Japan, 3 Innovative and Clinical Research Promotion Center, Gifu University Hospital, Yanagido, Gifu, Japan

* dskoike@gmail.com

## Abstract

### Introduction

Non-technical skills are essential for surgical patient safety and are implemented in clinical practice. However, training for non-technical skills has not been thoroughly investigated. This study aimed to evaluate the learning curve for non-technical skill-based education in herniorrhaphy.

### Methods

Quality improvement initiatives, including non-technical skill-based intervention, were performed in the department of surgery. The intervention included declaring the patient safety policy, briefing and debriefing, and criterion for the switching of places of the trainee and instructor as defined by the department. Patients who underwent herniorrhaphy from April 2014 to September 2017 were included.

### Results

A total of 14 trainees and nine instructors in the pre-intervention period and 14 trainees and seven instructors in the intervention period were included in this study. The median experience of each trainee was 28 and 15 cases in the pre-intervention and intervention groups, respectively. A total of 749 patients were included: 473 in the pre-intervention period and 328 in the intervention period. Demographics and hernia types were mostly similar between groups, and morbidity was not statistically different between the two groups (3.4 vs. 1.2%, p = 0.054). The nonlinear regression model showed an early decline and deep plateau phase of the learning curve in the intervention group. A significant difference was observed in the plateau operation time (61 min in the pre-intervention group and 52 min in the intervention group).

**Data Availability Statement:** All relevant data are within the manuscript and its Supporting Information files.

**Funding:** The author(s) received no specific funding for this work.

**Competing interests:** The authors have declared that no competing interests exist.

## Conclusion

This study demonstrated the effectiveness of non-technical skill-based intervention for surgical training. An early decline and deep plateau of the learning curve can be achieved with well-implemented quality improvement initiatives. Nonetheless, further studies are needed to establish a training program for non-technical skill-based learning.

## Introduction

Technical skills and new procedures have long been major confounders in modifying surgical outcomes. However, this is insufficient to improve surgical outcomes, even when technical skills are enhanced [1]. Non-technical skills (NTS) have recently been developed to improve surgical team performance. They are currently being developed for high-quality surgical procedures and applied in robotic surgery [2,3]. NTS is defined as "cognitive, social, and personal resource skills that complement technical skills and contribute to safe and efficient task performance." [4] NTS includes several concepts of cognitive, social, and personal skills, such as situational awareness, decision-making, communication and teamwork, and leadership. Moreover, they are now essential in surgical education and patient safety because of their effectiveness in improving surgical outcomes. NTS was developed and investigated as a scoring system in multiple fields and has already been implemented in surgical educational programs in the United States [5–7]. NTS can be learned using an appropriate training program [8].

Surgical education is important not only to improve surgical care and patient safety but also to instruct trainees in surgical procedures because surgeries within the learning curve have significantly poor outcomes, especially herniorrhaphy [9]. The learning curve of herniorrhaphy changes not only with experience, supervision of the instructor, and types of procedures but also with standardization and structured training programs [9,10]. However, the effectiveness of NTS on the surgical learning curve has not been sufficiently investigated. When NTS was performed intraoperatively, the surgical team helped the trainee with teamwork and decision-making. Trainees may not perform tasks solely during surgery. This may spoil the trainee's motivation or reduce the chance of training.

In our department, quality improvement initiatives were implemented with the patient safety approach policy, thereby improving patient outcomes through these initiatives [11]. In these initiatives, the instructor took turns with the operator or trainee in difficult cases to ensure patient safety. This study aimed to evaluate the educational effect of NTS-based education by analyzing the learning curve of herniorrhaphy, which is the first training procedure in a general surgery training program.

## Method

### Study setting and design

In this retrospective, single-center, cohort study, consecutive herniorrhaphy data between April 2014 and September 2017 at the Asahi General Hospital (AGH) were included. All the trainees in the AGH group were included in the study. Patients were excluded if they were younger than 18 years. Patients who had undergone emergency or elective surgeries were also included. If herniorrhaphy was performed with any procedure, only the time of herniorrhaphy was recorded. In cases of incarceration with bowel resection, data on operation time were excluded; however, data on surgeon experience were included. Simultaneous operation for bilateral hernia was calculated in two cases.

In the AGH, herniorrhaphy is performed by instant pairs of surgical trainees and instructors, and sometimes with other assistants. Trainees took a three-year educational program in the AGH, as specified by the Japanese Surgical Society (i.e., the curriculum for board-certified surgeons consisted of three years of education for a surgeon's trainee). All herniorrhaphy procedures performed during the study period included conventional surgeries and did not include minimally invasive procedures. The study was divided into two periods: pre-intervention period (PRE) from April 2014 to March 2016 and intervention period (INT) from April 2016 to September 2017. The learning curve for both periods was calculated based on the number of herniorrhaphies experienced by all surgeons.

## Intervention through education with NTS

The NTS intervention has been reported previously [11]. We used a bundle intervention approach, including the Non-Technical Skills for Surgeons (NOTSS) concept of the NTS, in quality improvement initiatives (Fig 1) [6]. The bundles included several strategies for improving operation time according to the NOTSS components. To enhance NTS, an instant pair of trainees and instructors performed briefings and debriefings for each surgery. These were performed outside the operative theater and included reviewing the case and confirming the procedure, even if it was a routine and standard surgery. The surgical team performed preoperative briefings to confirm the surgical strategy and procedures. The intervention used a grading system, and the main target operation time was set to 60 min. Strategies for the surgical team were developed according to the four grades and criteria for the switching of places of the trainee and instructor. Grade A was defined as < 40 min. Surgery was performed within a fair amount of time despite trainee participation. Grade B was defined as between 40 and 60 min. Surgery was performed within the standard time, and the instructor performed some of the procedures during the surgery. Grade C was defined as between 60 and 80 min. Surgery was performed over an excessive amount of time, and the instructor changed the operator to prevent further risks to patient safety. Grade D was defined as > 80 min. Surgery was performed for an excessive amount of time, and the instructor changed the operator and called for support from other surgeons. To distribute and implement the intervention, the strategies according to the operative time were displayed on a bulletin board and discussed in regular

| Situation awareness | Clearly defined grading system of operation time (pre-operative)<br>Shared information of the case with preoperative briefing (pre-operative) |
| --- | --- |
| Decision making | Option of surgical strategy according to the taking turns by instructor (intra-operative) |
| Communication and teamwork | Briefing of trainee and instructor for each surgery (pre-operative)<br>Shared target defined in the grading system (pre-operative)<br>Support for trainee to operation according to the briefing (intra-operative)<br>Debriefing of trainee and instructor for each surgery (post-operative) |
| Leadership | Standard procedure of herniorrhaphy defined in the department (pre-operative) |

**Fig 1. Relationship of the intervention and non-technical skills.**

weekly conferences involving all trainees and instructors in our office, as reported previously [11]. These strategies have been well implemented by surgical departments, attendants, and trainees.

## Data collection

Operative data were collected during quality improvement initiatives, and patient data were collected retrospectively. The operation time for herniorrhaphy was defined as the skin-to-skin. Patient age, sex, body mass index (BMI), hernia type, morbidity, and postoperative hospital stay were retrospectively analyzed. A giant hernia was defined as one that reached the scrotum. Patients with previous herniorrhaphies were defined as having recurrent hernias. Surgeon experience was counted only in the study institution, regardless of their experience in the previous institution. Hernia types were classified according to classifications published by the Japanese Hernia Society [12].

## Statistical analysis

We described patient characteristics using medians and interquartile ranges (IQRs) for continuous variables and frequencies for categorical variables. The Mann–Whitney U test was used to compare the two groups for continuous variables, and the chi-square test or Fisher's exact test was used for categorical variables. A nonlinear regression model was used to compare the association between operative time and surgical experience by group. The explanatory variables in the model included the interaction terms for experience and group (experience × group). Age, BMI, sex, hernia type, giant hernia, and recurrent hernia were included in the model to adjust for confounding factors. These variables were selected a priori based on discussions between the principal investigator and statistician. Restricted cubic spline curves were used to represent the nonlinear relationship between operation time and experience. To avoid overfitting, the number of knots was set to three, which was the least restrictive. Prediction curves for the association between operative time and experience, when the adjusted factor was fixed at the median, are presented for each group. Statistical analyses were performed using IBM SPSS version 25 (IBM Corp., Armonk, N.Y., USA) and R version 4.2.2 (The R Project for Statistical Computing). Two-sided $p < 0.05$ was considered statistically significant.

## Ethical approval

The ethics board committee of AGH approved this retrospective study (No. 2017112119). The requirement for informed consent from each patient was waived by the board committee because this study was part of quality improvement initiatives.

## Results

### Patient characteristics

A total of 809 patients underwent herniorrhaphy between April 2014 and September 2017. Based on our criteria, 42 and 17 patients in the PRE and INT groups, respectively, were excluded. A total of 749 patients were included in this study. Patient characteristics (Table 1) were comparable between the two groups, although the frequency of exclusion surgery was lower in the INT group than in the PRE group. There were no statistically significant differences in patient sex, age, BMI, or type of hernia between the two groups.

**Table 1. Demographic data of the patients.**

| | PRE group (n = 443) | INT group (n = 306) | P value |
|---|---|---|---|
| Gender, n (%) | | | 0.216 |
| Male | 407 (91.9) | 273 (89.2) | |
| Female | 36 (8.1) | 33 (10.8) | |
| Age, median (IQR) | 69 (62–77) | 70 (63–78) | 0.235 |
| BMI, median (IQR) | 23.1 (21.3–25.4) | 22.9 (21.2–24.6) | 0.374 |
| Laterality, n (%) | | | 0.885 |
| R | 214 (48.3) | 152 (49.7) | |
| L | 199 (44.9) | 132 (43.1) | |
| Bi | 30 (6.8) | 22 (7.2) | |
| Classification: JHS, n (%) | | | 0.463 |
| L | 378 (85.3) | 253 (82.7) | |
| M | 79 (17.8) | 59 (19.3) | |
| F | 13 (2.9) | 11 (3.6) | |
| others | 3 (0.7) | 5 (1.6) | |

Mann–Whitney U test, chi-square test.

PRE, pre-intervention period; INT, intervention period; JHS; Japanese Hernia Society.

## Trainees and instructors

A total of 33 surgeons were included in this study. In the PRE period, 14 trainees and nine instructors performed herniorrhaphy, while in the INT period, 14 trainees and seven instructors performed herniorrhaphy. The median experience case of each trainee was 28 [IQR 5.5–55] in the PRE period and 15 [IQR 7.25–39.25] in the INT period, which was statistically not significant (p = 0.401). The trainees in the first year of the surgical training program, which corresponded to post-graduate year 3, performed more than half of the procedures in both periods, 280 (59.2%) in the PRE period and 269 (82.0%) in the INT period, respectively.

## Patient outcome

The number of procedures performed was 473 in the PRE group and 328 in the INT group; bilateral hernias were counted twice (Table 2). The median operation time was 63.0 [IQR 51–80] min in the PRE group and 60.0 [IQR 45–75] min in the INT group, which were statistically different (p = 0.010). All procedures were performed using an open technique with mesh repair. The types of herniorrhaphy were mainly the modified Kugel patch method and the plug method and were not different in both groups. The median postoperative hospital stay was similar: 1.0 [IQR 1–1] in the PRE group and 1.0 [IQR 1–1] in the INT group. However, the mean postoperative hospital stay was 1.38 in the PRE group and 1.19 in the INT group, respectively, which were significantly different (p = 0.049). More day surgeries were performed in the INT group than in the PRE group (p = 0.019), because day surgery was implemented in our institution during the study period. Surgical complications occurred in 16 (3.4%) cases in the PRE group and in four (1.2%) cases in the INT group. The complication rates were not statistically different between the two groups (p = 0.054).

## Learning curves in each group with nonlinear regression analysis

The learning curve calculated using the nonlinear regression model is shown in Fig 2. The operation times between the two groups were significantly different after adjusting for

**Table 2. Surgical results of the patients.**

|  | PRE group | INT group | P value |
|---|---|---|---|
| Cases of herniorrhaphy | 473 | 328 |  |
| Recurrence hernia, n (%) | 16 (3.4) | 15 (4.6) | 0.39 |
| Giant hernia, n (%) | 54 (11.4) | 30 (9.1) | 0.302 |
| Operation time, median (IQR) | 63.0 (51–80) | 60.0 (47–75) | 0.010* |
| Hospital stay, median (IQR) | 1.0 (1–1) | 1.0 (1–1) | 0.049* |
| Day surgery, n (%) | 144 (30.4) | 125 (38.1) | 0.019* |
| Type of procedure |  |  | 0.697 |
| Modified Kugel | 322 (68.1) | 218 (66.5) |  |
| Plug | 148 (31.3) | 109 (33.2) |  |
| Others | 4 (0.8) | 1 (0.3) |  |
| Complication, n (%) |  |  |  |
| Bleeding | 2 (0.4) | 1 (0.3) | 0.791 |
| Infection | 0 (0) | 0 (0) | - |
| Organ injury | 2 (0.4) | 0 (0) | 0.239 |
| Recurrence | 1 (0.2) | 1 (0.3) | 0.792 |
| Other | 11 (2.3) | 2 (0.6) | 0.059 |
| Any | 16 (3.4) | 4 (1.2) | 0.054 |

Mann–Whitney test, X square test, *p < 0.05.

PRE, pre-intervention period; INT, intervention period.

covariates (β: -8.405, 95% confidence interval [CI] -15.015 to -1.796, p = 0.001). The learning curve showed a significant difference between the two groups after 15 cases, which continued until 58 cases. Over 58 experiences, 95% CI was greater because of fewer cases in both groups in that area. The interaction between experience and group was not significant (p < 0.535). Specifically, there was a constant reduction in operative time between the groups, regardless of the experience level. In both groups, a plateau was observed in 40 cases, and there was a 10-min significant difference in the plateau area between the two groups.

## Multivariable linear regression analysis for operation time (Table 3)

Univariable analysis showed that the study period, surgeon experience, BMI, male sex, giant hernia, and recurrent hernia were significant predictors of operation time. However, patient age and hernia type did not show significant coefficients. In the multivariable linear regression analysis, the study period, surgical experience, BMI, and giant and recurrent hernias demonstrated statistically significant coefficients for operation time.

## Discussion

This study demonstrated the impact of NTS on the learning curve for herniorrhaphy. Multivariable linear regression analysis showed that both surgical experience and NTS strategy were significant coefficient factors. The operation time during the learning curve decreased faster and greater in the INT group than in the PRE group, reflecting NTS implementation.

The shortening time effect on the learning curve was evident in this study. A decline in learning curves was observed in 30 or 40 cases and was stable in over 40 cases in both groups. A previous study showed that the learning curve of open or laparoscopic herniorrhaphy was similar to that in this study, with 30 cases required to stabilize the operation time for trainees

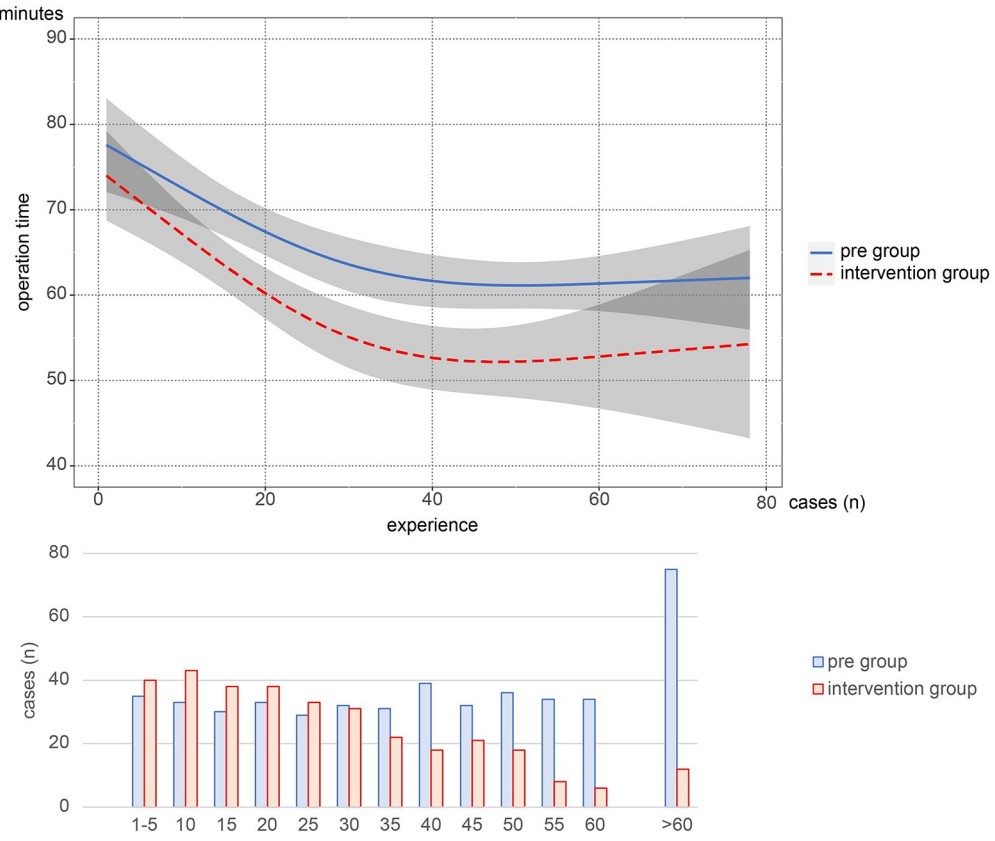

**Fig 2. Learning curve of herniorrhaphy using the nonlinear regression model.**

[13,14]. Although some studies showed that the progression of surgical skills continued in over 100 cases of herniorrhaphy, the process of surgical education for the novice phase in this study would be reliable for analysis [15,16].

**Table 3. Univariable and Multivariable linear regression analysis for operation time.**

| Variable | Univariable analysis | | | Multivariable analysis | | |
|---|---|---|---|---|---|---|
| | Coefficient | 95% CI | p-value | Coefficient | 95% CI | p-value |
| Experience | -6.911 | [-11.147, -2.676] | < 0.001* | -9.263 | [-13.447, -5.078] | < 0.001* |
| Period | -4.962 | [-10.096, 0.172] | 0.058 | -8.405 | [-15.015, -1.796] | 0.001* |
| Age | -0.097 | [-0.207, 0.012] | 0.081 | -0.048 | [-0.144, 0.048] | 0.33 |
| BMI | 1.9 | [1.24, 2.561] | < 0.001* | 1.678 | [1.075, 2.28] | < 0.001* |
| Gender | -9.822 | [-15.25, -4.394] | < 0.001* | -4.995 | [-9.992, 0.002] | 0.05 |
| Giant hernia | 22.141 | [12.57, 31.712] | < 0.001* | 21.061 | [12.29, 29.833] | < 0.001* |
| Recurrent hernia | 18.008 | [10.244, 25.772] | < 0.001* | 21.893 | [12.091, 31.695] | < 0.001* |
| Hernia type | | | | | | |
| L | - | - | 0.867 | - | - | 0.52 |
| M | -0.221 | [-4.038, 3.597] | | -1.448 | [-4.921, 2.026] | |
| F | -5.265 | [-17.891, 7.361] | | 5.539 | [-3.738, 14.815] | |
| O | 0.81 | [-10.776, 12.396] | | 1.392 | [-11.345, 14.13] | |

*p < 0.05.

Comparing the two groups, the operation time in the INT group was shorter than that in the PRE group in any learning phase. The strategies in this study included the changed policy of the operator; therefore, the difference in operation time in the early stage must be the subsequent change's effect. However, a faster decline in operation time could indicate that the trainee was able to perform the surgical technique better with less experience. In this study, the trainees in the PRE period should have more time to manipulate the surgical instruments and experience more trials and failures without changing policies and with longer operation times. By contrast, the trainees in the INT period should have more time to watch and learn the instructor's technique and experience fewer trials and greater success. A previous study has shown the importance of instructor supervision; however, the methodology or policy for supervision has not been investigated [9]. Training with NTS has irreplaceable potential for the learning curve, and a successful experience would be more effective in acquiring surgical techniques than failure.

The plateau phase is important in surgical training. The difference in operation time in the plateau phase was shown using nonlinear regression analysis. Surgical skills improved with experience, and a plateau was observed in certain cases in both groups. A previous study showed that the learning curve has several shapes for each surgeon and may have multiple plateau phases [15,17]. The various learning curves suggest that some surgeons experience early and shallow plateaus. The deep plateau time in the INT group could indicate the overcoming of the shallow plateau due to the intervention. Although the reason for this has not been clearly shown in this study, setting the target and intervention with NTS possibly moved the plateau to a deeper zone. The target and experience of success within the target time can enhance the educational effect on trainees.

Briefing and debriefing effects were advantageous for the INT group. The learning model showed that effective learning requires not only concrete experience but also active experimentation, reflective observation, and abstract conceptualization [18]. Education using NTS should enhance the learning model cycle. Procedures after briefing should play the role of active experimentation even if it is a routine and standard procedure because trainees have a risk of insufficient preparation for surgeries due to the duty task [19]. Only short preoperative communication, such as a surgical safety checklist, improves intraoperative communication [20]. Briefings in this study improved intraoperative communication and enhanced concrete experience. Debriefing helped with reflective observation and abstract conceptualization [21]. Unfortunately, the methodology of the briefing and debriefing was not standardized in this study. A recent study showed that policy change improves the debriefing quality [22]. Further studies are required to establish and improve the effectiveness of briefings and debriefings in surgical education.

## Limitation

This study has some limitations. First, the effectiveness of NTS in surgical training was clearly demonstrated; however, the generalizability of our results may be limited. This is because herniorrhaphy is the first procedure for surgical trainees. Although NTS have great potential for quality and safety in surgery, their effects on other procedures are unclear. Further studies are warranted to establish the educational effects of NTS for an entire surgical training. Second, the NTS of the operation team were not measured, while the entire policy, including NTS, was well-implemented in this study. An NTS scoring system, such as the NOTSS, requires a real-time investigator for measurement. Since NOTSS is not used in our clinical practice, any NTS-related competency of the trainee or instructor that might affect the results could not be measured. Further studies are essential to investigate the relationship between NTS scores and the

educational effects of NTS. There are also limitations owing to the retrospective, single-center nature of the study. Because the sample size was not calculated, it was unclear whether the number of cases and trainees was appropriate. This single-center study has the potential for selection bias in patients and trainees. A multicenter study is warranted to establish the effectiveness of NTS-based surgical education.

## Conclusion

This study demonstrates the effectiveness of an NTS-based intervention for surgical training. NTS-based training has the potential to improve surgical outcomes by facilitating the faster learning of trainees regarding operative time in the early phase. The deep plateau phase in the learning curve suggests that NTS-based out-of-theater training could overcome the initial barrier to surgical training without specialized resources. Further studies are warranted to establish a training program for NTS-based learning and its effectiveness in surgical education.

## Supporting information

**S1 Dataset.**
(XLSX)

## Author Contributions

**Conceptualization:** Daisuke Koike, Takahiro Nishimura, Yusuke Suka, Motoki Nagai, Yukihiro Nomura.

**Data curation:** Daisuke Koike, Takahiro Nishimura, Yusuke Suka, Motoki Nagai, Yukihiro Nomura.

**Formal analysis:** Takuma Ishihara.

**Methodology:** Daisuke Koike, Takahiro Nishimura, Yusuke Suka, Motoki Nagai, Yukihiro Nomura.

**Supervision:** Hiroyuki Kato, Yukio Asano, Masahiro Ito, Satoshi Arakawa, Akihiko Horiguchi.

**Writing – original draft:** Daisuke Koike.

**Writing – review & editing:** Daisuke Koike, Takahiro Nishimura, Yusuke Suka, Motoki Nagai, Yukihiro Nomura, Hiroyuki Kato, Yukio Asano, Masahiro Ito, Satoshi Arakawa, Takuma Ishihara, Akihiko Horiguchi.

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
