## [Decision Letter · Decision Letter 0]

14 Aug 2023

PONE-D-23-14386

Effects of non-technical skill-based surgical education for trainees on herniorrhaphy outcomes

PLOS ONE

Dear Dr. Daisuke Koike,

Thank you for submitting your manuscript to PLOS ONE. After careful consideration, we feel that it has merit but does not fully meet PLOS ONE’s publication criteria as it currently stands. Therefore, we invite you to submit a revised version of the manuscript that addresses the points raised during the review process.

We look forward to receiving your revised manuscript.

Kind regards,

Marco Clementi, Assistant Professor

Academic Editor

PLOS ONE

Journal Requirements:

Additional Editor Comments:

Dear Daisuke Koike,

I applaud all the Authors on their hard work on this project. However, the manuscript requires major revision to be accepted. 

Check the data in the table and the statistics. Provide details on the distribution of interventions between trainees and instructors in the two groups. Provide details on the type of hernioplasty adopted. "Conventional surgery" is too generic to be accepted

Reviewers' comments:

Reviewer's Responses to Questions

**Comments to the Author**

1. Is the manuscript technically sound, and do the data support the conclusions?

Reviewer #1: Yes

Reviewer #2: Yes

2. Has the statistical analysis been performed appropriately and rigorously? 

Reviewer #1: Yes

Reviewer #2: Yes

3. Have the authors made all data underlying the findings in their manuscript fully available?

Reviewer #1: No

Reviewer #2: Yes

4. Is the manuscript presented in an intelligible fashion and written in standard English?

Reviewer #1: Yes

Reviewer #2: Yes

5. Review Comments to the Author

Reviewer #1: 1. the topic is not unique but worthy of researching

2. there are hundreds of papers in google scholar and Refseek about this topic since 2019

3. the title is attractive

4. the abstract is informative

5. the aim is clear

6. the KEYWORDS are good

7. lack of the abbreviations section

8. the introduction provide sufficient background information for readers in the immediate field to understand the problem/hypotheses

9. the text arrangement is good

10. the method section is clear

11. the depth of the academic material is good

12. the study design is good

13. The suitability and accuracy of questions is good

14. The research methodology is clear

15. The materials are suitable

16. the logic is clear

17. the paper is not novel

18. there are few grammatical errors in this article

19. the related concepts are introduced

20. the readability is sufficient

21. the results are good

22. all figures/tables are clear enough to summarize the results for presentation to the readers

23. all figures/tables are well referred to in the text

24. the theoretical analysis in this article is sufficient

25. the discussion of results from multiple angles is sufficient

26. the conclusion is good

27. the reference section contains too many old ref

28. please use (google scholar and Refseek) search engines then set it since 2019

29. the references are in order within the text

30. Bias is present

31. There is no conflict of interest with the author about this topic

32. Fund is mentioned

33. Ethical approval is mentioned

34. Conflict of interest is mentioned

35. Acknowledgement is mentioned

36. You can use my suggestions

My final decision is acceptable after minor revision

Reviewer #2: The authors present an interesting topic on the effects of non-technical skill-based surgical education for trainees on herniorraphy outcomes. The authors found that after implementing their surgical educational program, there were decreased operative times, decreased hospital stay, and changes in day surgeries but no differences in morbidity or mortality outcomes.

The median hospital stay in Table 2 appears inconsistent from what is stated in the text.

It appears that the intervention group in Table 2 had fewer day surgeries than the pre-intervention group. This seems contradictory to the authors' argument that the interventional group had improved outcomes. This data would indicate that the intervention group required more overnight stays following herniorraphy than the pre-intervention group.

When the authors discuss experience, they do so in number of cases. I would be interested in seeing the experience broken down by PGY level and experience of the instructor (assistant professor vs. associate professor vs. professor).

The type of herniorraphy being performed (laparoscopic vs. robotic vs open) should also be explored as there are significant technical differences in the minimally invasive techniques vs. open techniques. In addition, the utilization of mesh vs. tissue based repairs should also be included as these could all affect the procedure and technical abilities displayed.

In conclusion, this study is on par with the push for EPAs around surgical training programs within the United States, but would need to address these issues listed above before it is ready for publication. I applaud the authors on their hard work on this project.

6. PLOS authors have the option to publish the peer review history of their article (what does this mean?). If published, this will include your full peer review and any attached files.

Reviewer #1: **Yes: **hazim abdul rahman alhiti

Reviewer #2: No

---

## [Author Response · Author response to Decision Letter 0]

31 Aug 2023

We would like to express our appreciation to the editor and the reviewers for their insightful comments, which helped us improve our paper significantly.

Journal Requirements:

Response:

We revised the manuscript according to PLOS ONE’s style requirements.

Response:

Accordingly, the Data availability statement was not enough based on the journal requirements. We submitted the minimal data set as a supporting information file required to replicate all study findings reported in the manuscript.

Additional Editor Comments:

Dear Daisuke Koike,

I applaud all the Authors on their hard work on this project. However, the manuscript requires major revision to be accepted.

1. Check the data in the table and the statistics. 

Response:

We confirmed the table and the statistics with a statistician, and no statistical defect has been found. The table formatting was not suitable for publication, so we revised all tables.

2. Provide details on the distribution of interventions between trainees and instructors in the two groups.

Response:

Accordingly, the details on the distribution of interventions in the INT group were important for this study. We clarified this point in the “Intervention through education with NTS” section. In the PRE group, no intervention was implemented.

 “To distribute and implement the intervention, the strategies according to the operative time were displayed on a bulletin board and discussed in regular weekly conferences involving all trainees and instructors in our office, as reported previously [10]” (Lines 118 to 121)

Provide details on the type of hernioplasty adopted. "Conventional surgery" is too generic to be accepted.

Response:

We are grateful for the comment. Accordingly, “Conventional surgery” was too generic. We have clarified the procedures in the “Study setting and design, Patient outcome” section and Table 2.

“All herniorrhaphy procedures performed during the study period included conventional surgeries and did not include minimally invasive procedures.” (Lines 92 to 93)

“All procedures were performed using an open technique with mesh repair. The types of herniorrhaphy were mainly modified Kugel patch method and plug method and were not different in both groups.” (Lines 189 to 191)

Reviewers' comments:

Reviewer's Responses to Questions

Comments to the Author

1. Is the manuscript technically sound, and do the data support the conclusions?

Reviewer #1: Yes Reviewer #2: Yes

Response:

We appreciate your favorable comment.

2. Has the statistical analysis been performed appropriately and rigorously?

Reviewer #1: Yes Reviewer #2: Yes

Response:

We appreciate your favorable comment.

3. Have the authors made all data underlying the findings in their manuscript fully available?

The PLOS Data policy requires authors to make all data underlying the findings described in their manuscript fully available without restriction, with rare exception (please refer to the Data Availability Statement in the manuscript PDF file). The data should be provided as part of the manuscript or its supporting information or deposited to a public repository. For example, in addition to summary statistics, the data points behind means, medians and variance measures should be available. If there are restrictions on publicly sharing data—e.g. participant privacy or use of data from a third party—those must be specified.

Reviewer #1: No Reviewer #2: Yes

Response:

Accordingly, the Data availability statement was not enough per the journal's requirements. We submitted the minimal data set as a supporting information file required to replicate all study findings reported in the manuscript.

4. Is the manuscript presented in an intelligible fashion and written in standard English?

Reviewer #1: Yes Reviewer #2: Yes

Response:

We appreciate your favorable comment. We corrected some grammar mistakes.

5. Review Comments to the Author

Reviewer #1: 

1. the topic is not unique but worthy of researching

Response:

We appreciate your favorable comment.

2. there are hundreds of papers in google scholar and Refseek about this topic since 2019

Response:

Accordingly, some of the recent important studies related to our study were not referenced. We revised the manuscript and added references No.3, 21, and 22.

3. the title is attractive

4. the abstract is informative

5. the aim is clear

6. the KEYWORDS are good

Response:

We appreciate your favorable comment.

7. lack of the abbreviations section

Response:

An abbreviations section was not provided in the manuscript because the journal does not require it. The journal only mentions that the authors should “Define abbreviations upon first appearance in the text,” which we did.

8. the introduction provide sufficient background information for readers in the immediate field to understand the problem/hypotheses

9. the text arrangement is good

10. the method section is clear

11. the depth of the academic material is good

12. the study design is good

13. The suitability and accuracy of questions is good

14. The research methodology is clear

15. The materials are suitable

16. the logic is clear

Response:

We appreciate your favorable comment.

17. the paper is not novel

Response:

Thank you for your comment on the study overview. Due to several limitations, the results of this study may not contribute significant novel findings regarding non-technical skills in healthcare. However, the study demonstrates that non-technical skill-based education improved the operation time and competency of the trainees. This finding would be valuable for the readers of PLOS ONE. We made a revision in the discussion section on the novelty of the study.

“However, a faster decline in operation time could indicate that the trainee was able to perform the surgical technique better with less experience.” (Lines 250 to 251)

18. there are few grammatical errors in this article

19. the related concepts are introduced

20. the readability is sufficient

21. the results are good

22. all figures/tables are clear enough to summarize the results for presentation to the readers

23. all figures/tables are well referred to in the text

24. the theoretical analysis in this article is sufficient

25. the discussion of results from multiple angles is sufficient

26. the conclusion is good

Response:

We appreciate your favorable comment.

27. the reference section contains too many old ref

Response:

Accordingly, there were several old references. Reference 11 was removed because it was old and almost irrelevant to this manuscript.

28. please use (google scholar and Refseek) search engines then set it since 2019

Response:

Accordingly, some of the recent important studies related to our study were not referenced. We revised the manuscript in the Discussion section and added references No.3, 21, and 22.

“They are currently being developed for high-quality surgical procedures and applied in robotic surgery [2,3].” (Lines 50 to 52)

“Debriefing helped with reflective observation and abstract conceptualization [21].” (Lines 279 to 280 )

“A recent study showed that policy change improves the debriefing quality [22].” (Lines 281 to 282)

29. the references are in order within the text

Response:

We appreciate your favorable comment.

30. Bias is present

Response:

Accordingly, several selection biases were present in this study. These were described in the limitations section.

31. There is no conflict of interest with the author about this topic 

32. Fund is mentioned

33. Ethical approval is mentioned

34. Conflict of interest is mentioned

35. Acknowledgement is mentioned 

Response:

We appreciate your favorable comment.

36. You can use my suggestions

My final decision is acceptable after minor revision

Response:

We appreciate your favorable comment. We revised the manuscript according to the reviewers’ recommendations.

Reviewer #2: The authors present an interesting topic on the effects of non-technical skill-based surgical education for trainees on herniorraphy outcomes. The authors found that after implementing their surgical educational program, there were decreased operative times, decreased hospital stay, and changes in day surgeries but no differences in morbidity or mortality outcomes.

Response:

We appreciate your favorable comment.

The median hospital stay in Table 2 appears inconsistent from what is stated in the text.

Response:

Accordingly, there were different descriptions of hospital stay in the manuscript and in Table 2. The data in the manuscript were mean hospital stay and those in Table 2 were median. Since many of the postoperative hospital stays were one day, both group data in Table 2 were ‘1.0 [1-1]’. We have clarified this point in the Patient outcome section.

“The median postoperative hospital stay was similar: 1.0 [IQR 1–1] in the PRE group and 1.0 [IQR 1–1] in the INT group. However, the mean postoperative hospital stay was 1.38 in the PRE group and 1.19 in the INT group, respectively, which were significantly different (p = 0.049).” (Lines 191 to 195)

It appears that the intervention group in Table 2 had fewer day surgeries than the pre-intervention group. This seems contradictory to the authors' argument that the interventional group had improved outcomes. This data would indicate that the intervention group required more overnight stays following herniorraphy than the pre- intervention group.

Response:

We appreciate the reviewer's concerns on this point. However, we consider our original text to describe the consistent finding. Accordingly, the number of day surgeries in the intervention group was smaller than that in the pre-intervention group (125 and 144, respectively). However, the proportion of day surgeries in the intervention group was larger than in the pre-intervention group (38.1% and 30.4%, respectively). Thus, we would like to retain the original text.

When the authors discuss experience, they do so in number of cases. I would be interested in seeing the experience broken down by PGY level and experience of the instructor (assistant professor vs. associate professor vs. professor).

Response:

Accordingly, the PGY levels and experience of the instructor were also interesting factors. Since the majority of the procedures were performed by PGY 3, a proportion of the PGY 3 has been described in the “Trainees and instructors” section. The PGY has not been used in univariable and multivariable analysis, because of multicollinearity risk with the number of experiences.

“The trainees in the first year of the surgical training program, which corresponded to post-graduate year 3, performed more than half of the procedures in both periods, 280 (59.2%) in the PRE period and 269 (82.0%) in the INT period, respectively.” (Lines 180 to 183)

The instructor’s position could not be classified because the study’s hospital was not an academic institution. Unfortunately, the data on the instructor’s experience was absent in this study. We added the limitation related to the instructors’ competency. 

“any NTS-related competency of the trainee or instructor that might affect the results could not be measured.” (Lines 295 to 297)

The type of herniorraphy being performed (laparoscopic vs. robotic vs open) should also be explored as there are significant technical differences in the minimally invasive techniques vs. open techniques. In addition, the utilization of mesh vs. tissue based repairs should also be included as these could all affect the procedure and technical abilities displayed.

Response:

We are grateful for the comment. Accordingly, the types of procedures were not clear. All surgeries in this study were performed using open techniques with a mesh. We have clarified the procedures in the “Study setting and design, Patient outcome” section and Table 2.

“All herniorrhaphy procedures performed during the study period included conventional surgeries and did not include minimally invasive procedures.” (Lines 92 to 93)

“All procedures were performed using the open technique with mesh repair. The types of herniorrhaphy were mainly the modified Kugel patch method and the plug method and were not different in both groups.” (Lines 189 to 191)

In conclusion, this study is on par with the push for EPAs around surgical training programs within the United States, but would need to address these issues listed above before it is ready for publication. I applaud the authors on their hard work on this project.

Response:

We appreciate your encouraging comment. The manuscript has been improved per the reviewer’s important recommendations.

---

## [Editor Report · Decision Letter 1]

4 Sep 2023

Effects of non-technical skill-based surgical education for trainees on herniorrhaphy outcomes

PONE-D-23-14386R1

Dear Dr. Daisuke Koike

We’re pleased to inform you that your manuscript has been judged scientifically suitable for publication and will be formally accepted for publication once it meets all outstanding technical requirements.

Kind regards,

Marco Clementi, Assistant Professor

Academic Editor

PLOS ONE

---

## [Editor Report · Acceptance letter]

11 Sep 2023

PONE-D-23-14386R1 

Effects of non-technical skill-based surgical education for trainees on herniorrhaphy outcomes 

Dear Dr. Koike:

I'm pleased to inform you that your manuscript has been deemed suitable for publication in PLOS ONE. Congratulations! Your manuscript is now with our production department. 

Kind regards, 

on behalf of

Dr. Marco Clementi 

Academic Editor

PLOS ONE